# NEURAL APPROXIMATION OF AN AUTO-REGRESSIVE PROCESS THROUGH CONFIDENCE GUIDED SAMPLING

## ABSTRACT

We propose a generic confidence-based approximation that can be plugged in and simplify the auto-regressive generation process with a proved convergence. We first assume that the priors of future samples can be generated in an independently and identically distributed (i.i.d.) manner using an efficient predictor. Given the past samples and future priors, the mother AR model can post-process the priors while the accompanied confidence predictor decides whether the current sample needs a resampling or not. Thanks to the i.i.d. assumption, the post-processing can update each sample in a parallel way, which remarkably accelerates the mother model. Our experiments on different data domains including sequences and images show that the proposed method can successfully capture the complex structures of the data and generate the meaningful future samples with lower computational cost while preserving the sequential relationship of the data.

## 1 INTRODUCTION

The auto-regressive (AR) model, which infers and predicts the causal relationship between the previous and future samples in a sequential data, has been widely studied since the beginning of machine learning research. The recent advances of the auto-regressive model brought by the neural network have achieved impressive success in handling complex data including texts (Sutskever et al., 2011), audio signals (Vinyals et al., 2012; Tamamori et al., 2017; van den Oord et al., 2016a), and images (van den Oord et al., 2016b; Salimans et al., 2017).

It is well known that AR model can learn a tractable data distribution $p(\mathbf{x})$ and can be easily extended for both discrete and continuous data. Due to their nature, AR models have especially shown a good fit with a sequential data, such as voice generation (van den Oord et al., 2016a) and provide a stable training while they are free from the mode collapsing problem (van den Oord et al., 2016b). However, these models must infer each element $x_i$ of the data $\mathbf{x} = [x_1, x_2, \cdots, x_i, \cdots, x_N]$ in a serial manner, requiring $\mathcal{O}(N)$ times more than the other non-sequential estimators, which outputs $\mathbf{x}$ at once (Garnelo et al., 2018; Kim et al., 2019; Kingma & Welling, 2014; Goodfellow et al., 2014). Moreover, it is difficult to employ recent parallel computation because AR models always require a previous time step by definition. This mostly limits the use of the AR models in practice despite their advantages.

To resolve the problem, we introduce a new and generic approximation method, *Neural Auto-Regressive model Approximator (NARA)*, which can be easily plugged into any AR model. We show that NARA can reduce the generation complexity of AR models by relaxing an inevitable AR nature and enables AR models to employ the powerful parallelization techniques in the sequential data generation, which was difficult previously.

NARA consists of three modules; (1) a prior-sample predictor, (2) a confidence predictor, and (3) the original AR model. To relax the AR nature, given a set of past samples, we first assume that each sample of the future sequence can be generated in an independent and identical manner. Thanks to the i.i.d. assumption, using the first module of NARA, we can sample a series of future priors and these future priors are post-processed by the original AR model, generating a set of raw predictions. The confidence predictor evaluates the credibility of these raw samples and decide whether the model needs re-sampling or not. The confidence predictor plays an important role in that the approximation errors can be accumulated during the sequential AR generation process if the

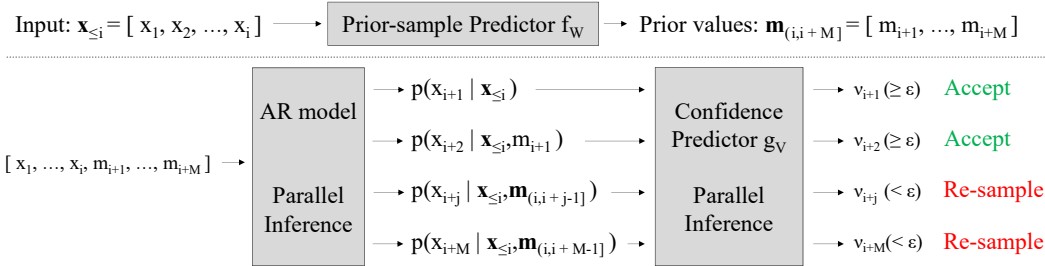

Figure 1: Conceptual description of Neural Auto-Regressive model Approximator (NARA). First, the prior-sample predictor $f_W$ predicts prior samples $\mathbf{m}$. Next, we draw $M$ samples from conditional probabilities for the given input $\mathbf{x}_{\leq i}$ and the prior samples $\mathbf{m}$ in parallel. Afterward, the confidence predictor $g_V$ estimates a confidence score $\nu_j$ of each $j$-th sample in parallel. Finally, we accept $k-1$ samples where $k = \arg\min_j \nu_j < \epsilon$. Here, $\epsilon$ is a predefined confidence threshold. More detailed figure is in Supplementary B.

erroneous samples with low confidence are left unchanged. Therefore, in our model, the sample can be drawn either by the mixture of the AR model or the proposed approximation method, and finally the selection of the generated samples are guided by the predicted confidence.

We evaluate NARA with various baseline AR models and data domains including simple curves, image sequences (Yoo et al., 2017), CelebA (Liu et al., 2015a), and ImageNet (Deng et al., 2009). For the sequential data (simple curves and golf), we employed the Long Short-Term Memory models (LSTM) (Hochreiter & Schmidhuber, 1997) as a baseline AR model while PixelCNN++ (Salimans et al., 2017) is used for the image generation (CelebA and ImageNet). Our experiments show that NARA can largely reduce the sample inference complexity even with a heavy and complex model on a difficult data domain such as image pixels.

The main contributions of our work can be summarized as follows: (1) we introduce a new and generic approximation method that can accelerate any AR generation procedure. (2) Compared to a full AR generation, the quality of approximated samples remains reliable by the accompanied confidence prediction model that measures the sample credibility. (3) Finally, we show that this is possible because, under a mild condition, the approximated samples from our method can eventually converge toward the true future sample. Thus, our method can effectively reduce the generation complexity of the AR model by partially substituting it with the simple i.i.d. model.

## 2 PRELIMINARY: AUTO-REGRESSIVE MODELS

Auto-regressive generation model is a probabilistic model to assign a probability $p(\mathbf{x})$ of the data $\mathbf{x}$ including $n$ samples. This method considers the data $\mathbf{x}$ as a sequence $\{x_i \mid i = 1, \cdots n\}$, and the probability $p(\mathbf{x})$ is defined by an AR manner as follows:

$$p(\mathbf{x}) = \prod_{i=1}^{n} p(x_i | x_1, ..., x_{i-1}), \tag{1}$$

From the formulation, the AR model provides a tractable data distribution $p(\mathbf{x})$. Recently, in training the model parameters using the training samples $\hat{\mathbf{x}}_T$, the computation parallelization are actively employed for calculating the distance between the real sample $\hat{x}_t \in \hat{\mathbf{x}}_T$ and the generated sample $x_t$ from equation (1). Still, for generating the future samples, it requires $\mathcal{O}(N)$ by definition.

## 3 PROPOSED METHOD

### 3.1 OVERVIEW

Figure 9 shows the concept of the proposed approximator *NARA*. *NARA* consists of a *prior-sample predictor* $f_W$ and *confidence predictor* $g_V$. Given samples $\mathbf{x}_{\leq i} = \{x_1, \cdots, x_i\}$, the prior-sample predictor predicts a chunk of $M$ number of the prior values $\mathbf{m}_{(i,i+M]}$. Afterward, using the prior samples, we draw the future samples $\mathbf{x}_{(i,i+M]}$ in parallel. We note that this is possible because the

prior $\mathbf{m}_{(i,i+M]}$ is i.i.d. variable from our assumption. Subsequently, for the predicted $\mathbf{x}_{(i,i+M]}$, the confidence predictor predicts confidence scores $\nu_{(i,i+M]}$. Then, using the predicted confidence, our model decides whether the samples of interest should be redrawn by the AR model (re-sample $x_{i+1}$) or they are just *accepted*. The detailed explanation will be described in the following sections.

## 3.2 APPROXIMATING SAMPLE DISTRIBUTION OF AR MODEL

Given the samples $\mathbf{x}_{\leq i} = \{x_1, \cdots, x_i\}$, a AR model defines the distribution of future samples $\mathbf{x}_{(i,j]} = \{x_{i+1}, \cdots, x_j\}$ as follows:

$$p_\theta(\mathbf{x}_{(i,j]}|\mathbf{x}_{\leq i}) = \prod_{l=i+1}^{j} p_\theta(x_l|x_1, ..., x_{l-1}). \tag{2}$$

Here, $\theta$ denotes the parameters of the AR model. The indices $i, j$ are assumed to satisfy the condition $j > i, \ \forall i, j \in [1, N]$. To approximate the distribution $p_\theta(\mathbf{x}_{(i,j]}|\mathbf{x}_{\leq i})$, we introduce a set of *prior* samples $\mathbf{m}_{(i,j-1]} = f_W(\mathbf{x}_{\leq i}; W)$, where we assume that they are i.i.d. given the observation $\mathbf{x}_{\leq i}$. Here, $W$ is the model parameter of the *prior-sample predictor* $f_W(\cdot)$.

Based on this, we define an approximated distribution $q_{\theta,W}(\mathbf{x}_{(i,j-1]}|\mathbf{x}_{\leq i}, \mathbf{m}_{(i,j-1]})$ characterized by the original AR model $p_\theta$ and the prior-sample predictor $f_W(\cdot)$ as follows:

$$
\begin{aligned}
q_{\theta,W}(\mathbf{x}_{(i,j]}|\mathbf{x}_{\leq i}, \mathbf{m}_{(i,j-1]}) &\equiv p_\theta(x_{i+1}|\mathbf{x}_{\leq i}) \underbrace{\prod_{l=i+2}^{j} p_\theta(x_l|\mathbf{x}_{\leq i}, m_{i+1}, \dots m_{l-1})}_{\text{Compute in parallel (const time)}} \\
&\overset{(A)}{\simeq} p_\theta(x_{i+1}|\mathbf{x}_{\leq i}) \underbrace{\prod_{l=i+2}^{j} p_\theta(x_l|\mathbf{x}_{\leq i}, x_{i+1}, \dots x_{l-1})}_{\text{Compute in sequential (linear time)}} \\
&= p_\theta(\mathbf{x}_{(i,j]}|\mathbf{x}_{\leq i}).
\end{aligned}
\tag{3}
$$

Here, approximation (A) is true when $\mathbf{m}_{(i,j-1]}$ approaches to $\mathbf{x}_{(i,j-1]}$. Note that it becomes possible to compute $q_{\theta,W}$ in a constant time because we assume the prior variable $m_i$ to be i.i.d. while $p_\theta$ requires a linear time complexity.

Then, we optimize the network parameters $\theta$ and $W$ by minimizing the negative log-likelihood (NLL) of $q_{\theta,W}(\mathbf{x}_{(i,j]} = \hat{\mathbf{x}}_{(i,j]}|\mathbf{x}_{\leq i}, f_W(\mathbf{x}_{\leq i}))$ where $\hat{\mathbf{x}}_{(i,j]}$ is a set of samples that are sampled from the baseline AR model. We guide the prior-sample predictor $f_W(\cdot)$ to generate the prior samples that is likely to come from the distribution of original AR model $\mathbf{m}$ by minimizing the original AR model $\theta$ and prior-sample predictor $W$ jointly as follows:

$$\min_{\theta, W} -\log p_\theta(\mathbf{x}_{(i,j]}^{(g)}|\mathbf{x}_{\leq i}) - \mathbb{E}_{p_\theta(\mathbf{x}_{(i,j]}|\mathbf{x}_{\leq i})}[\log q_{\theta,W}(\mathbf{x}_{(i,j]}|\mathbf{x}_{\leq i}, f_W(\mathbf{x}_{\leq i}))], \tag{4}$$

where $\mathbf{x}^{(g)}$ denotes the ground truth sample value in the generated region of the training samples. Note that both $p_\theta(x)$ and its approximated distribution $q_{\theta,W}(x)$ approaches to the true data distribution when (1) our prior-sample predictor generates the prior samples $\mathbf{m}$ close to the true samples $\mathbf{x}$ and (2) the NLL of the AR distribution approaches to the data distribution. Based on our analysis and experiments, we later show that our model can satisfy these conditions theoretically and empirically in the following sections.

## 3.3 CONFIDENCE PREDICTION

Using the prior-sample predictor $f_W(\cdot)$, our model generates future samples based on the previous samples. However, accumulation of approximation errors in the AR generation may lead to an unsuccessful sample generation. To mitigate the problem, we introduce an auxiliary module that determines whether to accept or reject the approximated samples generated as described in the previous subsection, referred to as *confidence predictor*.

First, we define the confidence of the generated samples as follows:

$$\nu_k = q_{\theta,W}(x_k = \hat{x}_k | \mathbf{x}_{\leq i}, f_W(\mathbf{x}_{\leq i})), \tag{5}$$

where $\hat{x}_k \sim p_\theta(x_k | \mathbf{x}_{\leq k-1})$ and $k \in \{1, \cdots, j\}$. The confidence value $\nu_k$ provides a measure of how likely the generated samples from $q_{\theta,W}(\cdot)$ is drawn from $p_\theta(x_k | \mathbf{x}_{\leq k-1})$. Based on the confidence value $\nu_k$, our model decides whether it can accept the sample $x_k$ or not. More specifically, we choose a threshold $\epsilon \in [0,1]$ and accept samples which have the confidence score larger than the threshold $\epsilon$.

When the case $\epsilon = 1$, our model always redraws the sample using the AR model no matter how our confidence is high. Note that our model becomes equivalent to the target AR model when $\epsilon = 1$. When $\epsilon = 0$, our model always accepts the approximated samples. In practice, we accept $\hat{k}(\epsilon) - 1$ samples among approximated $M$ samples where $\hat{k}(\epsilon) = \arg\min_k \nu_k > \epsilon$. Subsequently, we re-sample $x_{\hat{k}(\epsilon))}$ from the original AR model and repeat approximation scheme until reach the maximum length.

However, it is impractical to calculate equation (5) directly because we need the samples $\hat{x}_k$ from the original AR model. We first need to go forward using the AR model to see the next sample and come backward to calculate the confidence to decide whether we use the sample or not, which is nonsense.

To detour this problem, we introduce a network $g_V(\cdot)$ that approximates the binary decision variable $h_k^\epsilon = \mathbb{I}(\nu_k \geq \epsilon)$ as follows:

$$\mathbf{h}_{(i,j]}^\epsilon \simeq g_V(\mathbf{x}_{\leq i}, f_W(\mathbf{x}_{\leq i})), \tag{6}$$

where $\mathbf{h}_{(i,j]}^\epsilon = \{h_{i+1}^\epsilon, \cdots, h_j^\epsilon\}$. The network $g_V(\cdot)$ is implemented by a auto-encoder architecture with a sigmoid activation output that makes the equation (6) equivalent to the logistic regression.

## 3.4 Training Details

To train the proposed model, we randomly select the sample $s^{(i)} \in [1, N]$ for the sequence $\mathbf{x}^{(i)}$ in a training batch. Then, we predict $l^{(i)} = \min(B, N - s^{(i)})$ sample values after $s^{(i)}$, where $B$ denotes the number of samples the prediction considers. To calculate equation (4), we minimize the loss of the training sample $\mathbf{x}^{(k)}$, $k = 1 \cdots K$, and the locations $s^{(i)} \in [1, N]$ as,

$$\min_{\theta, W} -\frac{1}{K}\sum_{i=1}^{K} \log p_\theta(\mathbf{x}_{\leq s^{(i)}+l^{(i)}}^{(i)}) - \frac{1}{KM}\sum_{i=1}^{K}\sum_{j=1}^{M} \log q_{\theta,W}(\hat{\mathbf{x}}_{(s^{(i)}, s^{(i)}+l^{(i)}]}^{(i)(j)} | \mathbf{x}_{\leq s^{(i)}}^{(i)}). \tag{7}$$

Here, $\hat{\mathbf{x}}_{(s^{(i)}, s^{(i)}+l^{(i)}]}^{(i)(j)}$ for $j \in \{1 \cdots M\}$ denotes $M$ number of the sequences from the AR distribution $p_\theta(\mathbf{x}_{(s^{(i)}, s^{(i)}+l^{(i)}]} | \mathbf{x}_{\leq s^{(i)}}^{(i)})$ for $i$-th training data. From the experiment, we found that $M = 1$ sample is enough to train the model. This training scheme guides the distribution drawn by NARA to fit the original AR distribution as well as to generate future samples, simultaneously. To train $g_V(\cdot)$, binary cross-entropy loss is used with $\mathbf{h}^\epsilon$ in equation (6), with freezing the other parameters.

## 3.5 Theoretical Explanation

Here, we show that the proposed NARA is a regularized version of the original AR model. At the extremum, the approximated sample distribution from NARA is equivalent to that of the original AR model. In NARA, our approximate distribution $q(\mathbf{x}_{(i,j+1]} | \mathbf{x}_{\leq i}, \mathbf{m}_{(i,j]})$ is reformulated as follows:

$$q_{\phi,\theta}(\mathbf{x}_{(i,j+1]} | \mathbf{x}_{\leq i}, \mathbf{m}_{(i,j]}) \equiv p_\theta(x_{i+1} | \mathbf{x}_{\leq i}) \prod_{l=i+2}^{j+1} q_\phi(x_l | \mathbf{x}_{\leq i}, m_{i+1}, ..., m_{l-1})$$

$$= \underbrace{\frac{p_\theta(\mathbf{x}_{\leq i+1})}{p_\theta(\mathbf{x}_{\leq i})} \cdot \frac{p_\theta(\mathbf{x}_{\leq i+2})}{p_\theta(\mathbf{x}_{\leq i+1})}, \cdots, \frac{p_\theta(\mathbf{x}_{\leq j+1})}{p_\theta(\mathbf{x}_{\leq j})}}_{p_\theta(\mathbf{x}_{(i,j+1]} | \mathbf{x}_{\leq i})} \cdot \underbrace{\left[\frac{p_\theta(\mathbf{x}_{\leq i+1})}{q_\phi(\mathbf{x}_{\leq i}, m_{i+1})}, \cdots, \frac{q_\phi(\mathbf{x}_{\leq i}, \mathbf{m}_{(i,j]}, x_{j+1})}{p_\theta(\mathbf{x}_{\leq j+1})}\right]}_{\mathcal{R}(p_\theta, q_\phi, \mathbf{m}_{(i,j]})},$$

$$\tag{8}$$

where the parameter $\phi$ denotes the network parameters of the approximated distribution $q(\cdot)$. Therefore, our proposed cost function can be represented as the negative log-likelihood of the AR model with a regularizer $-\log\mathcal{R}(p_\theta, q_\phi, m_{(i,j]})$:

$$\min_{\phi,\theta} -\mathbb{E}_{p_\theta(\mathbf{x}_{(i,j+1]} | \mathbf{x}_{\leq i})}[\log p_\theta(\mathbf{x}_{(i,j+1]} | \mathbf{x}_{\leq i})) + \log\mathcal{R}(p_\theta, q_\phi, \mathbf{m}_{(i,j]})] \tag{9}$$

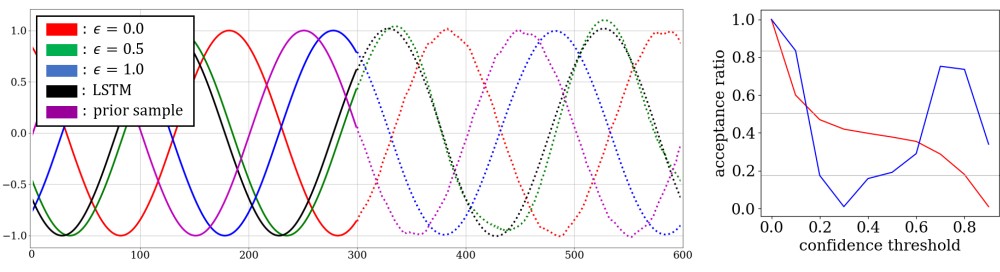

(a) One-dimensional time series data generation results.  (b) Error, accepts over $\epsilon \in [0.0, 1.0]$.

Figure 2: (a) One-dimensional time series data generation example. Dashed lines denote the generated samples. Red lines, green, and blue lines are from the approximation model with acceptance ratio $\epsilon = 0.0, 0.5$ and $\epsilon = 1.0$. Black line is the generation results from the baseline LSTM. Magenta line presents the generated prior samples. (b) The graph denoting the mean $\ell_1$ error (blue) and the acceptance ratio (red) over the threshold $\epsilon$.

Note that the proposed cost function is equivalent to that of the original AR model when $\log \mathcal{R}(p_\theta, q_\phi, \mathbf{m}_{(i,j)}) = 0$, which is true under the condition of $\mathbf{m}_{(i,j)} = \mathbf{x}_{(i,j)}$ and $q_\phi(\cdot|\cdot) = p_\theta(\cdot|\cdot)$. Here, $\mathbf{m}_{(i,j)} = f_W(\mathbf{x}_{\leq i})$. By minimizing the equation (9), $\mathcal{R}(p_\theta, q_\phi, \mathbf{m}_{(i,j)})$ enforces the direction of the optimization to estimate the probability ratio of $q_\phi$ and $p_\theta$ while it minimize the gap between $\frac{q_\phi(\mathbf{x}_{\leq i}, \mathbf{m}_{(i,j)}, x_{j+1})}{q_\phi(\mathbf{x}_{\leq i}, \mathbf{m}_{(i,j)})}$ so that $\mathbf{m}_{(i,j)} = f_W(\mathbf{x}_{\leq i})$ approaches to $\mathbf{x}_{(i,j)}$.

## 4 RELATED WORK

**Deep AR and regression models:** After employing the deep neural network, the AR models handling sequential data has achieved significant improvements in handling the various sequential data including text (Sutskever et al., 2011), sound (Vinyals et al., 2012; Tamamori et al., 2017), and images (van den Oord et al., 2016b; Salimans et al., 2017). The idea has been employed to "Flow based model" which uses auto-regressive sample flows (Kingma & Dhariwal, 2018; Germain et al., 2015; Papamakarios et al., 2017; Kingma et al., 2016) to infer complex distribution, and reported meaningful progresses. Also, the attempts (Yoo et al., 2017; Garnelo et al., 2018; Kim et al., 2019) to replace the kernel function of the stochastic regression and prediction processes to neural network has been proposed to deal with semi-supervised data not imposing an explicit sequential relationship.

**Approximated AR methods:** Reducing the complexity of the deep AR model has been explored by a number of studies, either targeting multiple domain (Seo et al., 2018; Stern et al., 2018) or specific target such as machine translation (Wang et al., 2018; Ghazvininejad et al., 2019; Welleck et al., 2019; Wang et al., 2018; 2019) and image generation (Ramachandran et al., 2017).

Adding one step further to the previous studies, we propose a new general approximation method for AR methods by assuming the i.i.d. condition for the "easy to predict" samples. This differentiates our approach to (Seo et al., 2018) in that we do not sequentially approximate the future samples by using a smaller AR model but use a chunk-wise predictor to approximate the samples at once. In addition, our confidence prediction module can be seen as a stochastic version of the verification step in (Stern et al., 2018), which helps our model to converge toward the original solution. This confidence guided approximation can be easily augmented to the other domain specific AR approximation methods because our method is not limited to a domain specific selection queues such as quotation (Welleck et al., 2019; Ghazvininejad et al., 2019) or nearby convolutional features (Ramachandran et al., 2017).

## 5 EXPERIMENTS

In this section, we demonstrate the data generation results from the proposed NARA. To check the feasibility, we first test our method into **time-series data generation problem**, and second, into **image generation**. The detailed model structures and additional results are attached in the Supplementary material. The implementation of the methods will be available soon.

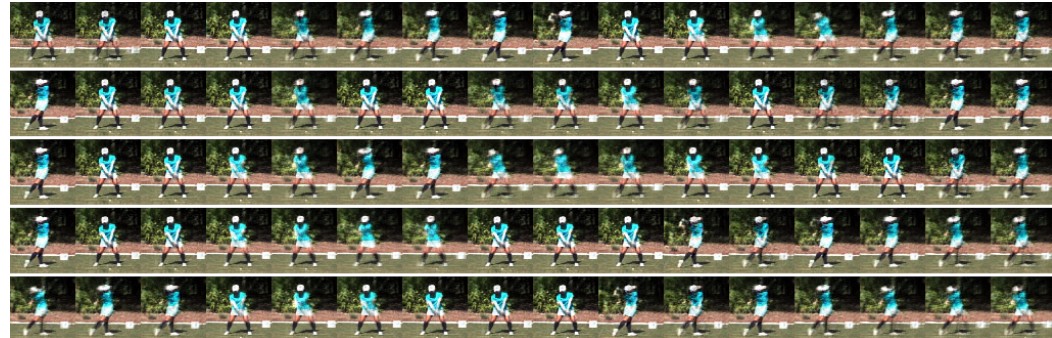

Figure 3: Image sequence generation examples (cropped in an equivalent time-interval). From the top to the bottom, each row denotes the generation result with $\epsilon \in \{1.00, 0.75, 0.50, 0.25, 0.00\}$.

Table 1: Mean $\ell_1$-error and the acceptance ratio over $\epsilon$ in image sequence generation.

| $\epsilon$ | 1.0 | 0.9 | 0.8 | 0.7 | 0.6 | 0.5 | 0.4 | 0.3 | 0.2 | 0.1 | 0.0 |
|---|---|---|---|---|---|---|---|---|---|---|---|
| Acceptance (%) | 0.0 | 15.3 | 43.0 | 59.3 | 69.1 | 77.1 | 82.1 | 84.5 | 88.0 | 92.5 | 100 |
| Mean error ($\ell_1$) | 21.1 | 26.8 | 24.9 | 21.7 | 18.4 | 19.9 | **18.3** | 20.6 | 19.4 | 25.3 | 24.3 |

## 5.1 EXPERIMENTAL SETTING

**Time-series data generation problem:** In this problem, we used LSTM as the base model. First, we tested our method with a simple one-dimensional sinusoidal function. Second, we tested the video sequence data (golf swing) for demonstrating the more complicated case. In this case, we repeated the swing sequences 20 times to make the periodic image sequences and resize each image to $64 \times 64$ resolution. Also, beside the LSTM, we used autoencoder structures to embed the images into latent space. The projected points for the image sequences are linked by LSTM, similar to (Yoo et al., 2017). For both cases, we used ADAM optimizer (Kingma & Ba, 2015) with a default setting and a learning rate 0.001.

**Image generation:** For the image generation task, we used PixelCNN++ (Salimans et al., 2017) as the base model. The number of channels of the network was set to 160 and the number of logistic mixtures was set to 10. See (Salimans et al., 2017) for the detailed explanation of the parameters. In this task, the baseline AR model (PixelCNN++) is much heavier than those used in the previous tasks. Here, we show that the proposed approximated model can significantly reduce the computational burden of the original AR model. The *prior-sample predictor* $f_W(\cdot)$ and the confidence estimator $g_V(\cdot)$ were both implemented by U-net structured autoencoder (Ronneberger et al., 2015). We optimized the models using ADAM with learning rate 0.0001. Every module was trained from scratch. We mainly used CelebA (Liu et al., 2015b) resizing the samples to $64 \times 64$ resolution. In the experiments, we randomly pick $36,000$ images for training and $3,000$ images for validation.

**Training and evaluation:** For the first problem, we use single GPU (NVIDIA Titan XP), and for the second problem, four to eight GPUs (NVIDIA Tesla P40) were used[1]. The training and inference code used in this section are implemented by PyTorch library. For the quantitative evaluation, we measure the error between the true future samples and the generated one, and also employ Fréchet Inception Distance score (FID) (Heusel et al., 2017) as a measure of the model performance and visual quality of the generated images for the second image generation problem.

## 5.2 ANALYSIS

### 5.2.1 TIME-SERIES DATA GENERATION

Figure 2a shows the generation results of the one-dimensional time-series from our approximation model with different acceptance ratios (red, green, and blue) and the baseline LSTM models (black).

---

[1]The overall expreiments were conducted on NSML (Sung et al., 2017) GPU system.

Figure 4: Generated images of $\epsilon \in \{0.0, 0.2, 0.4, 0.6, 0.8, 1.0\}$ on the CelebA dataset. The image in the left of the triplet denotes a accepted region (white: do accept, black: do not accept), the mid is a confidence map (red: high, blue: low) and the right shows the generated result.

From the figure, we can see that both models correctly generates the future samples. Please note that, from the prior sample generation result (magenta), the prior samples **m** converged to the true samples **x** as claimed in Section 3.5.

The graph in Figure 2b shows the acceptance ratio and the $\ell_1$-error over the confidence threshold $\epsilon \in (0, 1]$. The error denotes the distance between the ground truth samples **x** and the generated ones. As expected, our model accepted more samples as the threshold $\epsilon$ decreases. However, contrary to our initial expectations, the error-threshold graph shows that the less acceptance of samples does not always bring the more accurate generation results. From the graph, the generation with an intermediate acceptance ratio achieved the best result. Interestingly, we report that this tendency between the acceptance ratio and the generation quality was repeatedly observed in the other datasets as well.

Figure 3 shows the image sequence generation results from NARA. From the result, we can see that the proposed approximation method is still effective when the input data dimension becomes much larger and the AR model becomes more complicated. In the golf swing dataset, the proposed approximation model also succeeded to capture the periodic change of the image sequence. The table 3 shows that the proper amount of approximation can obtain better accuracy than the none, similar to the other previous experiments. One notable observation regarding the phenomenon is that the period of image sequence was slightly changed among different ratio of approximated sample acceptance (Figure 3). One possible explanation would be that the approximation module suppress the rapid change of the samples, and this affects the interval of a single cycle.

### 5.2.2 IMAGE GENERATION

Figure 4 shows that our method can be integrated into PixelCNN++ and generates images with the significant amount of the predicted sample acceptance (white region). We observed that the confidence was mostly low (blue) in the eyes, mouth, and boundary regions of the face, and the PixelCNN is used to generate those regions. This shows that compared to the other homogeneous regions of the image, the model finds it relatively hard to describe the details, which matches with our intuition.

The graphs in Figure 5 present the quantitative analysis regarding the inference time and the NLL in generating images. In Figure 5a, the relation between inference time and the skimming ratio is reported. The results show that the inference speed is significantly improved as more pixels are accepted. Table 2 further supports this that our approximation method generates a fair quality of images while it speeds up the generation procedure $5 \sim 10$ times faster than the base model.

In the image generation example also, we found that the fair amount of acceptance can improve the perceptual visual quality of the generated images compared to the vanilla PixelCNN++ (Table 2). Our method benefits from increasing the acceptance ratio to some extent in terms of FID showing a U-shaped trend over the $\epsilon$ variation, similar to those in Figure 2b. Note that a lower FID score identifies a better model. Consistent with previous results, we can conjecture that the proposed approximation scheme learns the mean-prior of the images and guides the AR model to prevent generating erroneous images. The confidence maps and the graph illustrated in Figure 4, 5a, and 5c

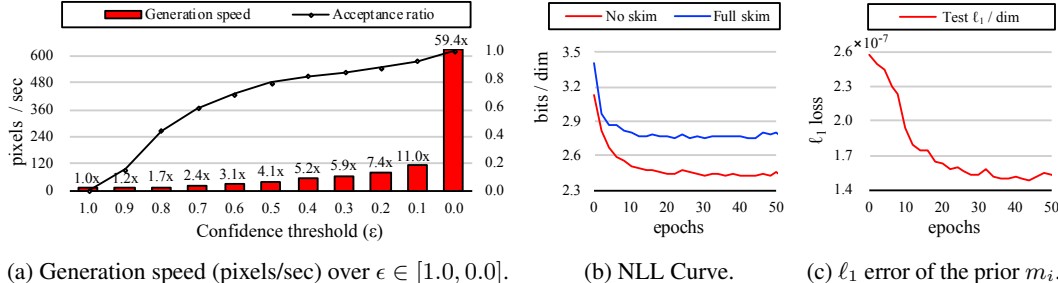

(a) Generation speed (pixels/sec) over $\epsilon \in [1.0, 0.0]$.     (b) NLL Curve.     (c) $\ell_1$ error of the prior $m_i$.

Figure 5: Quantitative analysis of the proposed method on CelebA dataset. (a) shows the generation speed over the threshold $\epsilon$. (b) shows the NLL of the inference with and without any acceptance. (c) presents the $\ell_1$-loss between the prior $\mathbf{m}$ and the pixel value $\mathbf{x}$ for every epoch.

Table 2: The speed up and visual quality of the results on CelebA dataset. The speed-up of the inference time with (SU) and without anchors (SU $\backslash$A) compared to that of PixelCNN++ is measured. acceptance ratio (AR) without anchors and FID (lower is better) are also displayed.

| $\epsilon$ | 1.0 | 0.9 | 0.8 | 0.7 | 0.6 | 0.5 | 0.4 | 0.3 | 0.2 | 0.1 | 0.0 |
|---|---|---|---|---|---|---|---|---|---|---|---|
| SU | 1.0x | 1.2x | 1.7x | 2.2x | 2.8x | 3.5x | 4.1x | 4.5x | 5.3x | 6.8x | 12.9x |
| SU $\backslash$A | 1.0x | 1.2x | 1.7x | 2.4x | 3.1x | 4.1x | 5.2x | 5.9x | 7.4x | 11.0x | 59.4x |
| AR (%) | 0.0 | 15.3 | 43.0 | 59.3 | 69.1 | 77.1 | 82.1 | 84.5 | 88.0 | 92.5 | 100 |
| FID | 56.8 | 50.1 | 45.1 | 42.5 | **41.6** | 42.3 | 43.4 | 45.9 | 48.7 | 53.9 | 85.0 |

support this conjecture. Complex details such as eyes, mouths and contours have largely harder than the backgrounds and remaining faces.

In Figure 5b and Figure 5c, the graphs show the results supporting the convergence of the proposed method. The graph in Figure 5b shows the NLL of the base PixelCNN++ and that of our proposed method under the full-accept case, i.e. we fully believe the approximation results. Note that the NLL of both cases converged and the PixelCNN achieved noticeably lower NLL compared to the fully accepting the pixels at every epoch. This is already expected in Section 3.2 that the baseline AR model approaches more closely to the data distribution than our module. This supports the necessity of re-generating procedure by using the PixelCNN++, especially when the approximation module finds the pixel has a low confidence.

The graph in Figure 5c presents the $\ell_1$ distance between the generated prior pixel $\mathbf{m}$ and the corresponding ground-truth pixel $\mathbf{x}$ in the test data reconstruction. Again, similar to the previous time-series experiments, the model successfully converged to the original value ($\mathbf{m}$ approaches to $\mathbf{x}$). Combined with the result in Figure 5b, this result supports the convergence conditions claimed in section 3.2. Regarding the convergence, we compared the NLL of the converged PixelCNN++ distribution from the proposed scheme and that of PixelCNN++ with CelebA dataset from the original paper (Salimans et al., 2017).

## 6 CONCLUSION

In this paper, we proposed the efficient neural auto-regressive model approximation method, NARA, which can be used in various auto-regressive (AR) models. By introducing the prior-sampling and confidence prediction modules, we showed that NARA can theoretically and empirically approximate the future samples under a relaxed causal relationships. This approximation simplifies the generation process and enables our model to use powerful parallelization techniques for the sample generation procedure. In the experiments, we showed that NARA can be successfully applied with different AR models in the various tasks from simple to complex time-series data and image pixel generation. These results support that the proposed method can introduce a way to use AR models in a more efficient manner.

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

## A    IMPLEMENTATION DETAILS

**Time serious sample generation:**  For the 1-dimensional samples case, the LSTM consists of two hidden layers, and the dimension of each layer was set to $51$. We observed $o = 200$ steps and predicted $p = 20$ future samples. The *chunk-wise predictor* $f_W(\cdot)$ and the *confidence predictor* $g_V(\cdot)$ were defined by single fully-connected layer with input size 200 and the output size 20. In this task, our model predicted every 20 samples by seeing the 200 previous samples.

For the visual sequence case, we used autoencoder structures to embed the images into latent space. The encoder consists of four "Conv-activation" plus one final Conv filters. Here, the term "Conv" denotes the convolutional filter. Similarly, the decoder consists of four "Conv transpose-activation" with one final Conv transpose filters. The channel sizes of the encoder and decoder were set to $\{32, 32, 32, 64\}$ and $\{64, 32, 32, 32\}$. All the Conv and Conv Transpose filters were defined by $3 \times 3$. The activation function was defined as Leaky ReLU function.

The embedding space was defined to have 10-dimensional space, and the model predicts every 5 future samples given 50 previous samples. The prior sample predictor and the confidence predictor are each defined by a fully-connected layers, and predict future samples in conjunction to the decoder. We note that the encoder and decoder were pre-trained by following (Kingma & Welling, 2014).

**Image generation:**  The prior sample and confidence predictors consist of an identical backbone network except the last block. The network consists of four number of "Conv-BN-activation" blocks, and the four number of "Conv transpose-BN-Activation" block. All the Conv and Conv Transpose filters were defined by $4 \times 4$. The last activation of the prior sample predictor is tangent hyperbolic function and that of confidence predictor is defined as sigmoid function. The other activation functions are defined as Leaky-ReLU. Also, the output channel number of the last block are set to 3 and 1, respectively. The channel size of each convolution filter and convolution-transpose filter was set to be $\{64, 128, 256, 512\}$ and $\{256, 128, 64, 3\}$.

We used batch normalization (Ioffe & Szegedy, 2015) for both networks and stride is set to two for all the filters. The decision threshold $\epsilon$ is set to the running mean of the $\kappa \log(q_{\theta,W}(\cdot))$. We set $\kappa = 2.5$ for every test and confirmed that it properly works for all the cases.

## B    SUPPLEMENTARY EXPLAIN ON PROPOSED SAMPLE GENERATION

Figure 6 shows the detailed process of sampling when the proposed NARA module is attached to the mother AR model. The diagram describes the Sinusoidal function generation example, which is the simplest example in our paper. In each auto-regressive step, we decide if we use the approximated value predicted by our sample predictor (predict the value in chunkwise manner assuming i.i.d condition) or sample the value with the mother auto-regressive model. This decision is conducted based on the confidence estimation by the Confidence predictor of the paper. The chunk-wise estimation step can be boosted by recent parallel computing methods different from the mother auto-regressive model, and this can possibly relax the computation burden of the original auto-regressive model. More importantly, this scheme can be applied to various auto-regressive models.

## C    SUPPLEMENTARY EXPERIMENTS

### C.1    SUPPLEMENTARY GENERATION RESULTS

In addition to the results presented in the paper, we show supplement generation examples in below figures. Figure 3 and Table 3 present the image sequence generation result from the other golf swing sequence. In this case also, we can observe the similar swing cycle period changes and acceptance ratio-error tendencies reported in the paper. Our approximation slightly affects the cycle of the time-serious data, and the result also shows that the approximation can achieve even better prediction results than "none-acceptance" case.

Figure 11 and 12 shows the additional facial image generation results among $\epsilon \in [0.0, 1.0]$. We can see that pixels from boundary region were more frequently re-sampled compared to other relatively simple face parts such as cheek or forehead. Also, we tested our model with ImageNet classes, and the results were presented in Figure 10.

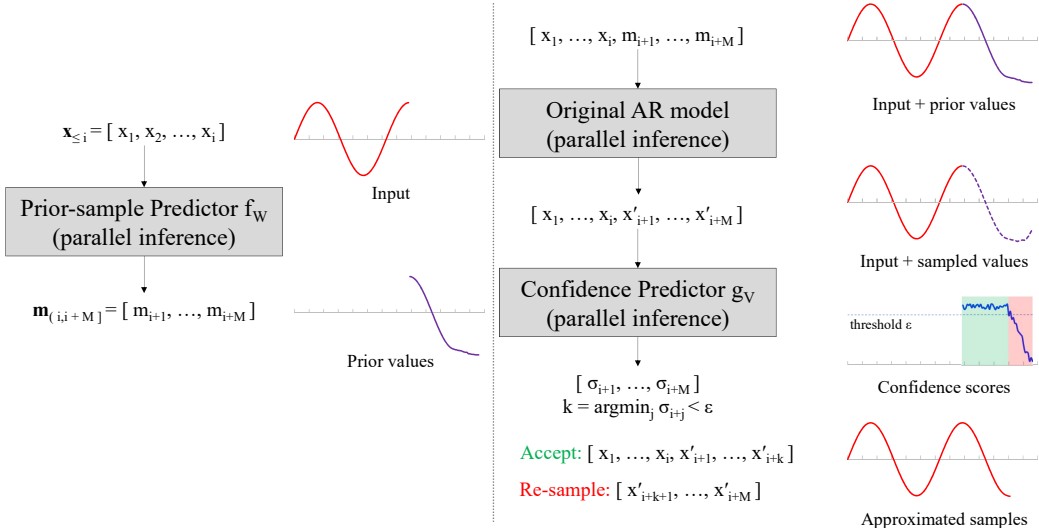

Figure 6: Detailed description of the sample generation process.

## C.2 ANALYSIS ON WHEN CONFIDENCE MODULE FAILS.

If the prior-sample predictor $f_W$ performs worse than our expectation, the confidence module $g_V$ will reject all prior samples; hence, in this case, the model repeats to draw samples using the original AR model in a sample-by-sample manner. However, in an even worse situation, the confidence module could always accept prior-samples, including low-quality prior samples. To simulate the failure or both $f_W$ and $g_V$, we manually fix the "accept region" regardless of the predicted confidence score. We select ten samples with the lowest confidence scores and report the results in Figure 8. In the figure, we can observe that the drastic error can occur when both the prior-sample predictor $f_W$ and confidence module $g_V$ fail. However, in practice, our confidence module $g_V$ can detect the drastic error during the sampling stage as shown in Figure 8b; hence, such extreme error will occur rarely.

## C.3 APPLYING NARA WITH FASTER PIXELCNN++

To show that our method can be added to diverse AR mode, we also combined our skimming method to the fast version of PixelCNN++[2]. Figure 9 and Table 4 show the generated samples and the generation time using the incorporated model (*skim*+Fast PixelCNN++). Due to the lack of time to fully investigate all datasets with the new implementation, we conducted the experiment on CIFAR-10, which is the most widely used dataset in PixelCNN works and is more diverse than the CelebA dataset. From the results, we show that our method, augmented with Fast PixelCNN++, can make the baseline algorithm much faster as we suggested in the paper.

Table 3: Mean $\ell_1$-error and the acceptance ratio over $\epsilon$ in image generation sequence.

| $\epsilon$ | 1.0 | 0.9 | 0.8 | 0.7 | 0.6 | 0.5 | 0.4 | 0.3 | 0.2 | 0.1 | 0.0 |
|---|---|---|---|---|---|---|---|---|---|---|---|
| Acceptance (%) | 0.0 | 18.0 | 35.0 | 44.5 | 56.1 | 61.0 | 70.5 | 85.5 | 92.5 | 99.0 | 100 |
| Mean Error | 2.20 | 2.61 | 2.21 | **1.79** | 2.28 | 2.10 | 1.91 | 1.87 | 1.94 | 2.38 | 2.45 |

---

[2]Ramachandran, Prajit, et al. "Fast generation for convolutional autoregressive models." arXiv preprint arXiv:1704.06001 (2017).

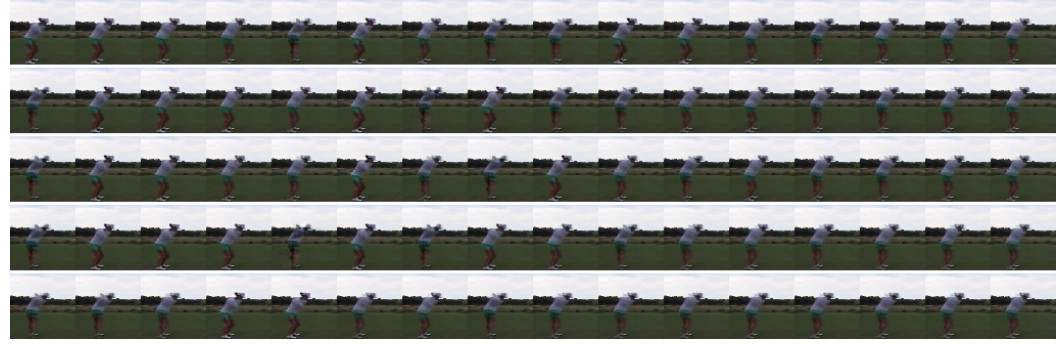

Figure 7: Image sequence generation examples (cropped in an equivalent time-interval). From the top to the bottom rows, each denotes the generation result with $\epsilon \in \{1.00, 0.75, 0.50, 0.25, 0.00\}$.

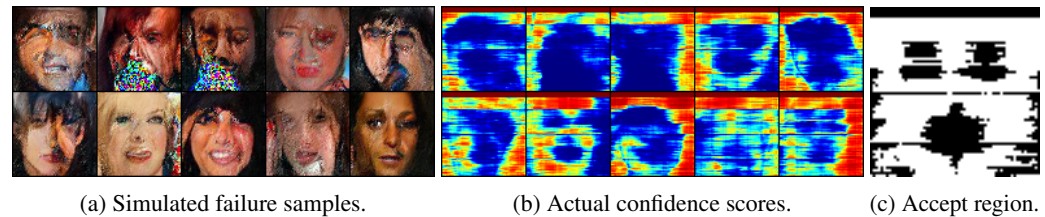

(a) Simulated failure samples.     (b) Actual confidence scores.     (c) Accept region.

Figure 8: Simulated failure samples in generating face images by fixing the "accept region" regardless of actual confidence scores. We report ten samples with lowest actual confidence scores.

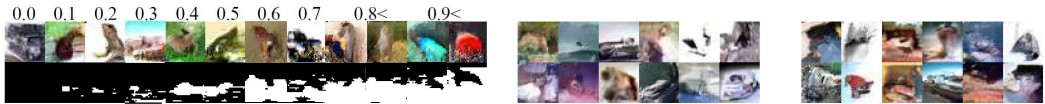

Figure 9: Generated CIFAR-10 images when combined with Fast PixelCNN++. Left (*NARA* + Fast PixelCNN++), center (Fast PixelCNN++), right (PixelCNN++). The numbers on top denote $1 - \epsilon$.

Table 4: Generation time per an image and skimming ratio over $\epsilon$ on CIFAR-10 Dataset. Here, **[F]** denotes the original Fast PixelCNN++. Note that the maximum acceptance ratio is $28/32 = 87.5\%$ since our method starts from the four lines which is already drawn by the original Fast PixelCNN++.

| $\epsilon$ (CIFAR-10) | 1.0 | 0.8 | 0.6 | 0.4 | 0.2 | 0.0 | [F] |
|---|---|---|---|---|---|---|---|
| Time (sec) | 24.2 | 23.2 | 19.6 | 16.1 | 12.5 | 4.87 | **23.8** |
| AR (%) | 0.00 | 11.0 | 25.4 | 40.7 | 55.5 | 87.5 | - |

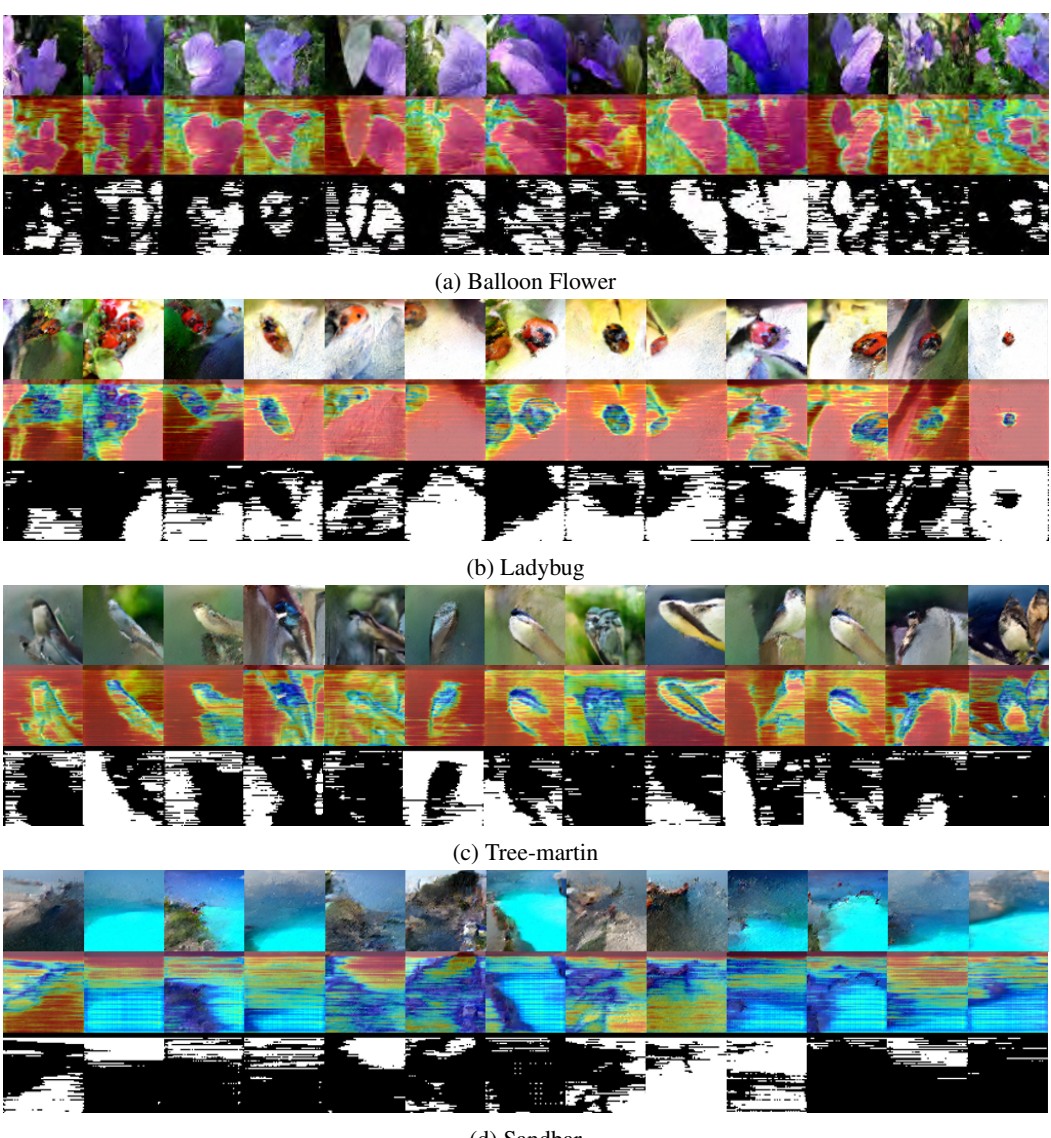

(a) Balloon Flower

(b) Ladybug

(c) Tree-martin

(d) Sandbar

Figure 10: ImageNet generation examples.

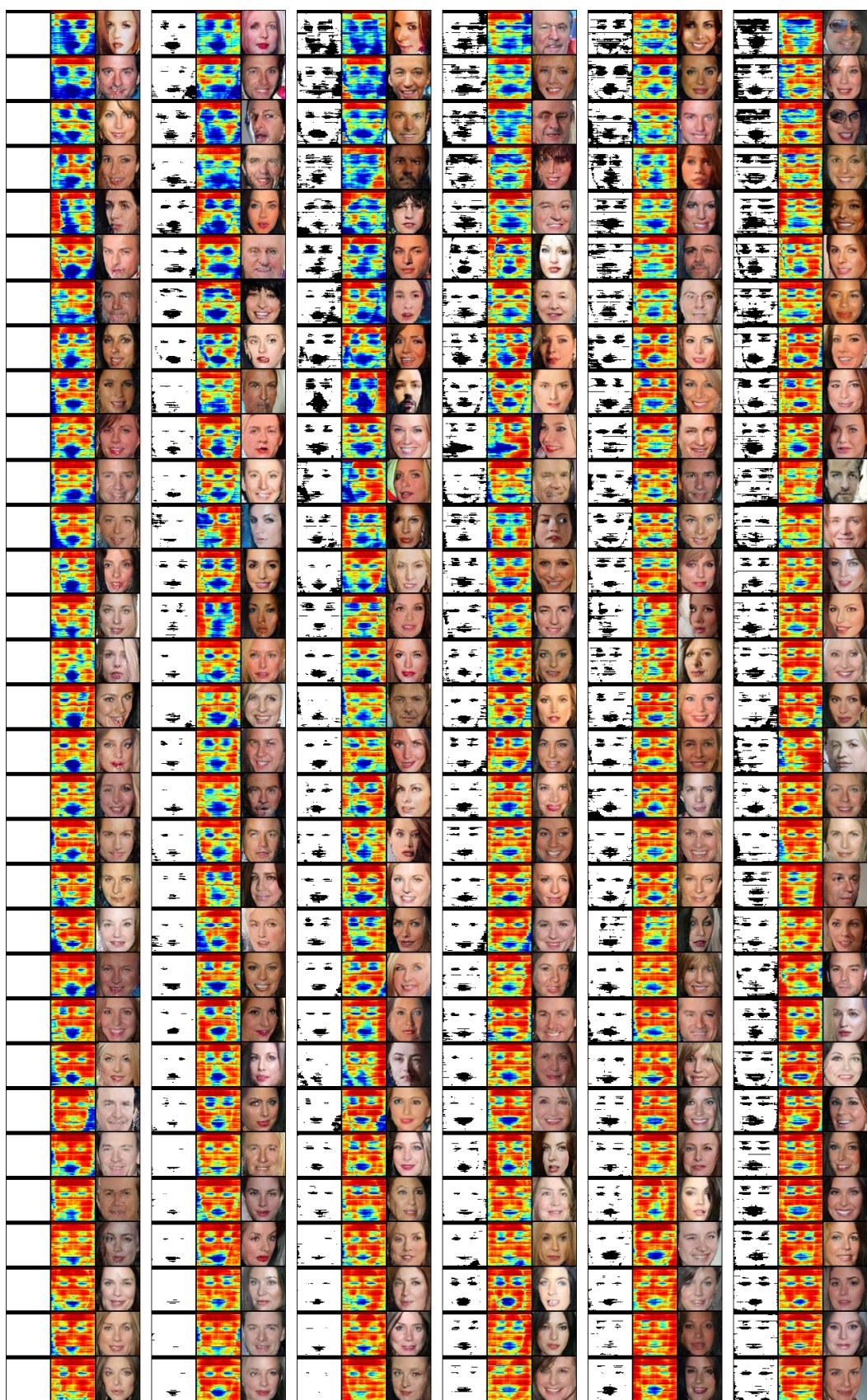

Figure 11: CelebA generation result in $64 \times 64$. $\epsilon \in \{0.0, 0.1, 0.2, 0.3, 0.4, 0.5\}$.

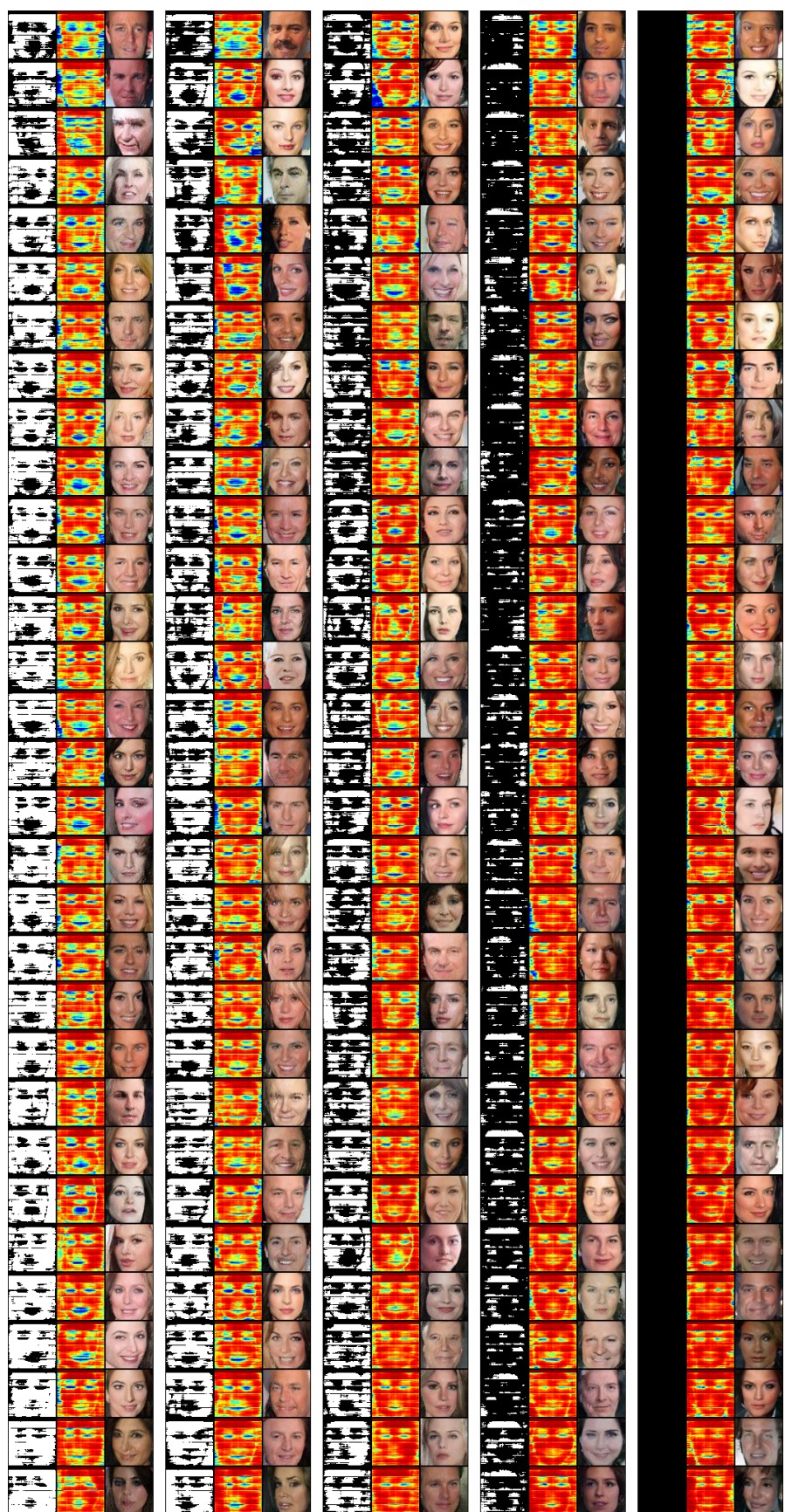

Figure 12: More selected CelebA generation result in $64 \times 64$. according to $\epsilon \in \{0.6, 0.7, 0.8, 0.9, 1.0\}$.

