# OpenReview forum: "Neural Approximation of an Auto-Regressive Process through Confidence Guided Sampling"
_ICLR.cc/2020/Conference — Reject_

### Official Review · AnonReviewer2 · 2019-10-19
**Official Blind Review #2**

**Rating:** 6

**Review:**

The paper presents a technique for approximately sampling from autoregressive models using something like a a proposal distribution and a critic. The idea is to chunk the output into blocks and, for each block, predict each element in the block independently from a proposal network, ask a critic network whether the block looks sensible and, if not, resampling the block using the autoregressive model itself.

In broad strokes the approach makes sense. It assumes, essentially, that parts of the sequence are hard to predict and parts are easy and, if there are enough easy parts, this procedure should lead to faster inference.

The paper's writing is not ideal. There are some grammatical mistakes that harm reading (for example, the second paragraph of the introduction says "However, these models must infer each element of the data x ∈ RN step by step in a serial manner, requiring O(N) times more than other non-sequential estimators", where it is unclear what is O(N) more than what, how is this measured, etc). That said I was mostly able to follow all key points.

The authors do not point out the obvious connection to GANs, which also rely on a critic network to decide whether a sample looks like it comes from the correct distribution, except in GANs the critic is jointly trained with the generator (as opposed to here where it's trained after) and in GANs the critic is only used at training time, while here the critic is used to accelerate sampling (the better the critic the faster this method can sample).

I wish the experimental results were a little more explicit about the time vs quality tradeoff; I expected to see more plots with pareto curves, since as-is it's hard to judge the magnitude of the tradeoffs involved. I'd also like a more thorough analysis on why there is a non-monotonic tradeoff in some experiments (table 1, figure 2(b)) between the amount of approximation and the sample quality; this makes me think something else is going on here as this approximate inference method should just decrease quality, never increase it.

Overall I lean towards accepting the paper, but I encourage the authors to revise the writing and to add a few plots explicitly showing the time vs quality tradeoff both in likelihood (wrt the full model) and in downstream metrics like FID.

**Experience Assessment:**

I have published one or two papers in this area.

**Review Assessment: Checking Correctness Of Derivations And Theory:**

I assessed the sensibility of the derivations and theory.

**Review Assessment: Checking Correctness Of Experiments:**

I assessed the sensibility of the experiments.

**Review Assessment: Thoroughness In Paper Reading:**

I read the paper thoroughly.

---

### Official Review · AnonReviewer3 · 2019-10-21
**Official Blind Review #3**

**Rating:** 3

**Review:**

The authors consider the problem of sampling time series.
To solve the problem they propose a method that is based on the autoregression model. The novelty here lies in the proposed sampling methods: we start with a sampling of a prior and then try to generate data according to the restored distribution. We learn two functions: signal recovery and confidence prediction.
The main hyperparameter of the algorithm $\varepsilon$ identifies how much samples we accept.
The distinctive feature of the algorithm is speed-up for the sample generation process.

Weak reject

There are a significant number of works on video generation, see e.g. [1, 2], references therein and articles that cite these two articles. The problem setting seems pretty similar. It seems like a good idea to compare to these methods (and it seems that video generation is a very resource-demanding procedure, and they don't use parallel applications similar to proposed in the paper. What is the reason?) Most of the approaches use only one frame to generate video, but it seems that LSTM in these methods will benefit from using of multiple frames as input (and will be able to transfer information in autoregression manner by transferring all they need in a hidden state).
The article, in my opinion, will benefit from comparison to these approaches or at least by using some benchmarks from these works to demonstrate feasibility of the considered approach, also it seems that these works are good for demonstration of parallelization capabilities (as in many cases the same idea applies).

Not minor comments:
1. In Figure 2 (a) it is not clear how the data and prediction were generated. According to the procedure in Figure 1 and text we use the same input for all approaches. However solid lines for different epsilons are different.
2. The effect of the dependence of recovery of quality for Figure 2 (b) is not explained and is controversial: we get the smallest error for intermediate acceptance ratio, however, there is also a decrease of error if we further increase the gauge threshold (btw the term gauge threshold is new to machine learning community, consider replacement of it)
3. More simpler examples will benefit the paper, as we'll be able to know more fundamental properties of the proposed approach.


Minor technical comments:
1. s. 3.1. predictor predicts
commas in equation (8)
2. Figure 2: no axis labels for the left plot, use for label "acceptance ratio" red color font & for label "L1 error" blue color font
3. Table 1 bracket after l_1 is missing
4. Maybe $\sigma$ is not the best designation of confidence, as it can be confused with the variance
5. Figure 1: some indexes should be not $x_{i + 2}$, but $x_{i + j}$, $x_{i + M}$. Also, some ">" before \epsilon should be "<"
6. "a auto-encoder architecture" ->
"an auto-encoder architecture"

[1] J. He et al. Probabilistic Video Generation using Holistic Attribute Control. 2018. ECCV
[2] E. Denton, R.Fergus. Stochastic Video Generation with a Learned Prior. 2018.

**Experience Assessment:**

I do not know much about this area.

**Review Assessment: Checking Correctness Of Derivations And Theory:**

I assessed the sensibility of the derivations and theory.

**Review Assessment: Checking Correctness Of Experiments:**

I carefully checked the experiments.

**Review Assessment: Thoroughness In Paper Reading:**

I read the paper thoroughly.

---

### Official Review · AnonReviewer1 · 2019-10-23
**Official Blind Review #1**

**Rating:** 6

**Review:**

This paper addresses the sequential limitation of autoregressive model when doing sampling. Specifically, instead of sampling next observations in a sequential fashion, this paper generates future observations in parallel, with the help of i.i.d. future priors. With the help of learned confidence model, the model is able to get trade-off between speed and approximation accuracy. Experiments on synthetic data and image generation with PixelCNN++ show the comparable results while being significantly faster than baseline.

Overall the paper is well motivated, with an interesting design of the variational distribution to approximate the true autoregressive distribution. The design of the confidence model looks a bit heuristic, but the trade-off ability between efficiency and quality it brings is also quite interesting.

Below are some minor comments:

1. The theoretical analysis is basically comment about the objective which is less interesting. However more interesting guarantees would be: 1) with the additional correction term added, how would it help with reducing the variance; 2) As the q_{\theta, \phi} is always in a limited form due to the parallelism requirement, how bad the approximation could be in the worst case ---- I’m not asking for these results, but any form of discussion would be helpful.

2. The author only compared with the raw PixelCNN++. Would any of the existing AR-speedup method be applicable for a comparison?


**Experience Assessment:**

I have published one or two papers in this area.

**Review Assessment: Checking Correctness Of Derivations And Theory:**

I assessed the sensibility of the derivations and theory.

**Review Assessment: Checking Correctness Of Experiments:**

I assessed the sensibility of the experiments.

**Review Assessment: Thoroughness In Paper Reading:**

N/A

---

### Decision · Program_Chairs · 2019-12-19

**Decision:**

Reject

**Comment:**

The paper presents a technique for approximately sampling from autoregressive models using something like a a proposal distribution and a critic. The idea is to chunk the output into blocks and, for each block, predict each element in the block independently from a proposal network, ask a critic network whether the block looks sensible and, if not, resampling the block using the autoregressive model itself.
The idea in the paper is interesting, but the paper would benefit from
- a better relation to existing methods
- a better experimental section, which details the hyper-parameters of the algorithm (and how they were chosen) and which provides error bars on all plots (and tables)